# Esketamine and Psilocybin—The Comparison of Two Mind-Altering Agents in Depression Treatment: Systematic Review

**DOI:** 10.3390/ijms231911450

**Published:** 2022-09-28

**Authors:** Dominika Psiuk, Emilia Magdalena Nowak, Natalia Dycha, Urszula Łopuszańska, Jacek Kurzepa, Marzena Samardakiewicz

**Affiliations:** 1Students Scientific Association at the Chair and Department of Psychology, Medical University of Lublin, 20-059 Lublin, Poland; 2Students Scientific Association at the Department of Medical Chemistry, Medical University of Lublin, 20-059 Lublin, Poland; 3Chair and Department of Psychology, Medical University of Lublin, 20-059 Lublin, Poland; 4Department of Medical Chemistry, Medical University of Lublin, 20-059 Lublin, Poland

**Keywords:** depression, psychedelics, psilocybin, esketamine

## Abstract

This publication discusses two compounds belonging to the psychoactive substances group which are studied in the context of depression treatment—psilocybin and esketamine. The former is a naturally occurring psychedelic. The latter was invented in the laboratory exactly 60 years ago. Although the substances were controversial in the past, recent studies indicate the potential of those substances as novel antidepressant agents. The PubMed/MEDLINE database was used to identify articles for systematic review, using the following search terms: (depression) AND (psilocybin) OR (ketamine). From 617 items, only 12 articles were obtained in the final analyses. Three articles were devoted to psilocybin in depression treatment and nine to esketamine. In most studies, esketamine showed a significant reduction in both depressive symptoms and suicidal ideation shortly after intake and after a month of treatment compared to baseline and to standard-of-care antidepressant agents. Psilocybin’s antidepressive effects occurred one day after intake and after 6–7 weeks of treatment and were maintained for up to 6 or 8 months of follow-up. One study indicated that psilocybin’s effects are comparable with and may be superior to escitalopram treatment. Both esketamine and psilocybin demonstrated rapid and long-term effects in reducing depression symptoms and, after overcoming some limitations, may be considered as novel antidepressant agents in future.

## 1. Introduction

Depression is one of the most common and debilitating mental conditions and affects more than 280 million people worldwide, which is an equivalent of 3.8 percent of the whole population [1]. It is a complex disorder with genetic, epigenetic, and environmental etiology, which involves anatomical and functional changes in brain development [2]. The pathophysiology of the disease has not been fully explained yet, but there are several possible mechanisms, including altered serotonergic, noradrenergic, dopaminergic, and glutamatergic systems, increased inflammation, hypothalamic–pituitary–adrenal (HPA) axis abnormalities, vascular changes, and decreased neurogenesis and neuroplasticity [3]. Depression is a serious health condition, especially when recurrent, treatment-resistant, and with moderate or severe intensity and can lead to a death by suicide [1,4]. Suicidal thoughts, behaviors, and intents are some of the symptoms of a depressive episode [1]. The suicide risk in patients with mood disorders has been estimated to be 10–30 times higher than in the general population [5]. The statistics indicate that over 700,000 people die by suicide every year and that depression remains one of its most common causes; similarly, psychological autopsy studies have repeatedly reported depression as the most common mental illness among suicide decedents [1,6].

### 1.1. Current Depression Treatments

The most common treatment strategies of the depressive disorder include pharmacological treatment, psychotherapy, and physical therapy (electroconvulsive therapy [ECT] and transcranial magnetic stimulation [TMS], among others) [7]. Psychological treatment usually consists of psychotherapy, e.g., cognitive–behavioral therapy (CBT), behavioral activation (BA) therapy, problem-solving therapy (PST), and interpersonal therapy (IPT). The basic antidepressant medications are selective serotonin-reuptake inhibitors (SSRIs) and tricyclic antidepressants (TCAs) [1]. Despite the moderate outcomes and its limitations, pharmacotherapy remains the first-line treatment for patients with moderate and severe depression [7]. The most important limitation of pharmacotherapy is treatment-resistant depression (TRD), defined as not achieving remission after two or more treatment trials of first-line antidepressants. TRD affects approximately 30 percent of people treated for a major depressive episode and is associated with increased severity and recurrence of disease, which results in prolonged function impairments, medical comorbidities, and increased suicidality as well as non-suicidal mortality [7,8]. Other consequences include a higher economic cost, caregiver burden, and associated stigma. Another challenge of the pharmacotherapy of depression is the long time to drug onset, which leaves a gap between the start of the treatment and its effects [5,9]. Data from placebo-controlled trials, as well as reports from professional experience, clearly indicate that during the first several weeks of antidepressant usage, suicidal thoughts and suicidal behavior in adolescents and young adults might increase. However, the risk of an actual suicide attempt or a suicide death is difficult to quantify [9,10].

The most common depression development theory refers to serotonin imbalance in brain tissue, which would explain the validity of the usage of selective serotonin-reuptake inhibitors in depression treatment; however, this concept has recently been suggested to be less accurate than initially assumed [10]. Therefore, there is a need to reconsider the current methods of treatment and introduce novel antidepressive agents.

### 1.2. The Potential of Mind-Altering Substances

For over 30 years now, mind-altering substances such as ketamine and psychedelics such as psilocybin, dimethyltryptamine (DMT), and d-lysergic acid diethylamide (LSD) have been studied in psychiatric disorder treatment [11,12]. However, the first psychedelic trials were conducted in the 1950s and continued until prohibited in the mid-1960s due to the sociopolitical issues with the rising popularity of their recreational use and despite the evidence of their effectiveness in the treatment of psychiatric conditions [13]. Ketamine, first synthesized in the 1960s, also has a history with misuse and was placed under federal restrictions in the late 1990s, though only a few years later researchers started to study its antidepressive effects [14,15]. In 2019, the U.S. Food and Drug Administration (FDA) approved esketamine nasal spray for treatment-resistant depression therapy [16].

Psilocybin (3-[2-(Dimethylamino)ethyl]-1H-indol-4-yl dihydrogen phosphate) is a natural substance present in Grophoriacees mushrooms, genus psilocybe [17]. Psilocybin is a prodrug that is rapidly dephosphorylated in vivo into psilocin, its psychoactive compound [18]. Psilocybin decreases cerebral blood flow in the amygdala, an area found to be hyperreactive in depression, and in the posterior cingulate cortex, which regulates emotion, memory, and consciousness [19]. The action onset occurs 20–40 min after per os intake of 0.2 mg/kg body mass and lasts about 4–6 h [20].

Psilocybin and psilocin are based on the tryptamine structure and have a configuration of oxygen and nitrogen atoms similar to that of serotonin, which allows them to bind and affect 5-HT2 receptors, particularly 5-HT2A and 5-HT2C [18,20]. This mechanism may underlie the antidepressant effect of psilocybin; however, psilocybin causes an additional effect, i.e., the experience of an altered state of consciousness that seems to have a therapeutic outcome as well, especially if used in conjunction with psychotherapy [21]. Psilocybin administration presents positive outcomes among patients with depression and anxiety resulting from suffering from terminal disease and is also studied in the treatment and prevention of cluster headaches [17].

Despite its mind-altering effects, which are also the reason behind the psilocybe mushrooms’ recreational use, psilocybin seems to be safe, particularly if administered in a controlled environment [22,23]. According to the Independent Scientific Committee on Drugs’ 2010 report, considered within 16 criteria (harm to users and harm to others), psilocybin mushrooms are the least harmful among the controlled substances in the United Kingdom, causing drug-specific and drug-related impairment of mental functioning far less dangerous than impairments from such drugs as alcohol, tobacco, or benzodiazepines [24]. Psilocybin seems not to lead to dependence, as serotonergic psychedelics do not directly affect the dopaminergic system [11]. Moreover, it does not impair liver function nor cause tissue toxicity [17]. The lethal dose, 50 percent (LD50) determined for rats is 280 mg/kg body mass by the intravenous route [25].

Esketamine ((2S)-2-(2-chlorophenyl)-2-(methylamino)cyclohexan-1-one) is a nonselective, noncompetitive S-enantiomer of racemic ketamine which presents a 3–4 times stronger affinity for the NMDA (N-methyl-D-Aspartic Acid) receptor than its R-enantiomer, arketamine [26,27,28,29]. As an NMDA antagonist, esketamine causes increased glutamate release which stimulates GLUR2 (glutamate receptor) and enhances neurotrophic signaling. As a result, it affects the brain functions which regulate emotions and mood [29]. One of the most common psychotomimetic effects connected with ketamine usage is dissociation [30]. According to the International Statistical Classification of Diseases and Related Health Problems (ICD-10) definition, dissociative disorders are a “partial or complete loss of the normal integration between memories of the past, awareness of identity and immediate sensations, and control of bodily movements” [31]. Dissociative anesthesia is a state after intravenous or intramuscular ketamine administration wherein patients appear awake, do not respond to sensory inputs, but preserve spontaneous respiratory activity [32]. Much research has shown that dissociation might also occur after esketamine usage, and is one of the most common adverse effects during the treatment of TRD [26,33,34,35,36,37,38,39,40]. The dissociative symptoms and psychoactive effects of ketamine can lead to increasing recreational usage, and rapidly developing tolerance might lead to an increased dosage over time [32].

Administered intranasally, esketamine is absorbed rapidly, with bioavailability at about 50 percent. Maximum plasma concentration is achieved 20–40 min after intake and the average terminal (T ½) is 4–12 h [28]. It is mainly metabolized to noresketamine in the liver via cytochrome P450 enzymes. Noresketamine is an active metabolite and shows weaker activity towards the NMDA receptors. The LD50 for ketamine hydrochloride is 447 mg/kg body mass for rats, by oral administration [28,29].

Esketamine has been used in TRD treatment since 2019, in combination with selective serotonin-reuptake inhibitors (SSRIs) or serotonin–norepinephrine reuptake inhibitors (SNRIs). Due to its adverse events, dissociation and sedation in particular, and the potential risk of abuse, its administration is supervised in certified clinics and strictly monitored by Risk Evaluation and Mitigation Strategy (REMS) [16,28,29].

Recent studies have proven that both esketamine and psilocybin can produce rapid effects that alleviate depressive symptoms. This review aims to evaluate these studies, compare these mind-altering substances and, if possible, assess their validity as potential antidepressant agents.

## 2. Methods

The aim of this work is to compare and review the potential use of two psychoactive substances, esketamine and psilocybin, in depression treatment. Based on the guidelines provided by the Primary Reporting Items for Systematic Reviews and Meta-Analyses Statement (PRISMA), the PubMed/MEDLINE database was used to identify potential articles for analysis using the following search terms: (depression) AND (psilocybin) OR (ketamine). The literature search was conducted on 30 December 2021, and 617 items were obtained.

The results were filtered for time (last 10 years, 590 results), character of the studies (clinical trials, 43 results), and participant type (human participants only, 43 results). The following types of studies were excluded: questionnaire-based surveys, surveys specifying the state of the respondents’ knowledge, meta-analyses, and reviews. Only studies that examined psilocybin and esketamine in patients with depression were included as the main focus of this review was to examine the influence of these substances on depressive symptoms. Moreover, only studies where esketamine was administered intranasally were included as it is an approved route.

Next, two people read the abstracts of the identified studies and excluded those that did not meet the selection criteria. As a result of the above searches, a total of 12 studies were included for review. All studies were randomized, controlled trials. The selection process is illustrated in Figure 1.

This systematic review has been registered in PROSPERO database: CRD42022351685.

## 3. Results

All of the 12 studies included in this review are presented in Table 1.

Within psilocybin studies, two compared its effects to placebo alone among patients with depression in life-threatening diseases [41,42]. One study compared psilocybin to escitalopram, a selective serotonin-reuptake inhibitor (SSRI), among patients with Major Depressive Disorder (MDD) [43]. All esketamine studies compared esketamine effects to standard-of-care antidepressants in Treatment Resistant Depression (TRD) [26,33,34,35,36,37,38,39,40].

### 3.1. Rapid Effects

Six studies comparing esketamine nasal spray in conjunction with oral antidepressant treatment to oral antidepressant alone indicated esketamine’s rapid effects [26,33,34,35,37,40], measured 2 to 4 and 24 h after esketamine intake (Table 2). Within the studies, there was a rapid decrease in depressive symptoms at both time points, measured with the Montgomery-Asberg Depression Rating Scale (MADRS), with a significant difference for the 84 mg esketamine dose [26,33,37,40] but not for flexible dosing, ranging from 56 to 84 mg. One study demonstrated a significant mean difference from the baseline and within groups 24 h after drug administration for 56 mg esketamine [26]. One study failed to assess statistical significance for rapid esketamine effects because the primary endpoint for this study was not met [34].

Psilocybin’s rapid onset was reported in only one study, conducted by Ross et al., 2016. A statistically significant reduction from the baseline in Hospital Anxiety and Depression Scale HADS for Depression (HADS D) and Beck’s Depression Inventory (BDI) was observed 1 day post-psilocybin administration, both in the first, pre-crossover, and the second, post-crossover sessions. There was a significant difference between the psilocybin-first and placebo-first groups 1 day after the pre-crossover session, measured in HADS D and BDI. One day after the second session, which was a crossover, no significance between the two groups was observed in BDI, as the psilocybin intake alleviated depressive symptoms in the placebo-first group, while the score in the psilocybin-first group did not change since the first session (about 6 weeks) [42].

### 3.2. Long-Term Outcomes

Esketamine was administered twice weekly in conjunction with the standard-of-care oral antidepressant therapy (duloxetine, escitalopram, sertraline, venlafaxine, paroxetine, or mirtazapine) taken daily for 2 [26] or 4 weeks [33,34,35,37,38,39,40]. A downward trend in depressive symptoms was observed on the 25th and 28th day of esketamine treatment (Table 3). Statistical significance was achieved in two studies [35,40] and could not be assessed in one study due to not meeting the primary endpoint [34]. One study managed to determine the median time to relapse during the post-esketamine treatment period, which was 34 days and 44 days for both remitters and responders, as well as for responders who were not in remission, respectively [39]. Another study assessed the number of relapses after 17.7–19.4 weeks of esketamine treatment; 24 and 16 relapses were observed among stable remitters and stable responders, respectively, versus 39 and 34 in the placebo group, respectively [36]. Response and remission rates at 4–7-week follow-up were, in general, higher within patients receiving esketamine than those who were administered only standard-of-care antidepressants (Table 4) [26,34,35,38,40].

Psilocybin was administered during two sessions, spaced 3–7 weeks apart, in counterbalanced sequence [41,42,43]. Additionally, in one study patients were receiving escitalopram daily for 6 weeks [43]. In all three studies, psilocybin demonstrated antidepressant effects 6–7 weeks after administration (Table 5). Griffiths, 2016, and Ross, 2016, also assessed that these effects were sustained up to 6- and 8-monts’ follow up. The response and remission rates are presented in Table 4 [41,42]. A recent study conducted by Carhart-Harris compared the efficacy of two psilocybin sessions with a 6-week SSRI (escitalopram) treatment. Although no significance between groups was observed, the study showed that psilocybin can be used as an antidepressant with comparable effects. Furthermore, secondary endpoints generally favored psilocybin over escitalopram, e.g., response and remission rates at a 6-week time point were 70% and 57% vs. 48% and 28% for psilocybin and escitalopram, respectively. The significance of the secondary endpoints could not be assessed [43].

### 3.3. Suicidality

In most of the esketamine studies, suicidal ideation or behavior were key exclusion criteria [26,34,35,36,38,39] but were an inclusion requirement in three studies [33,37,40]. Esketamine demonstrated significant, rapid improvement in MADRS suicidal thoughts or in Clinical Global Impression of Severity of Suicidality Scale (CGI-SS) score compared to placebo after 4 h [33,37]. The reduction was also observed 24 h after intake [33,37], and it reached significance in one study [40]. The analysis among subgroups suggested that esketamine treatment is effective among patients who had attempted suicide in the past and among patients with severe depressive symptoms [37,40]. The improvement in severity of suicidality was also observed on the 25th day of the treatment [37,40].

Nevertheless, suicidal thoughts occurred in some patients during esketamine treatment, regardless of the presence of initial suicidal symptoms [33,34,35,36,37,38,39,40]. Suicidal behaviors were significantly less frequent and arose mainly in the studies where suicidal ideation and behavior were inclusion criteria—during the double-blind phase in three studies [34,36,40] and during follow-up in two studies [37,40]. In some trials, suicidal ideation and suicidal attempts were reported as severe adverse events [33,36,37,39,40] and in one they were the cause of withdrawal [39]. However, in most studies, they were considered by investigators as doubtfully related or unrelated to esketamine [33,37,39]. The incidence of these symptoms, assessed by the Columbia Suicide Severity Rating Scale (C-SSRS) in esketamine groups, was approximately comparable to placebo groups [34,35,37,38,40].

In the case of psilocybin, its effects against suicidality were assessed in the Carhart-Harris study on the Suicidal Ideation Attributes Scale (SIDAS) at the sixth week of the study. The decrease in SIDAS scores was higher in the psilocybin group than in the escitalopram group and was −2.0 vs. −0.8 points from baseline [43].

### 3.4. Adverse Events

Adverse events (AEs) from the reviewed studies were summarized and are shown in Table 6 and Table 7. None of the psilocybin studies revealed any serious AE, whether medical or psychiatric, during the administration period. The most common psychiatric AEs were psychological discomfort and transient anxiety, while the most common somatic AEs were headaches and migraines, elevated blood pressure and heart rate, nausea, and vomiting. All AEs were transient, tolerable, and had resolved fully by the end of the sessions. Regarding psychiatric AEs, no pharmacological interventions (e.g., benzodiazepines, anti-psychotics) were needed during the psilocybin dosing sessions, no participants became addicted to psilocybin, there were no cases of prolonged psychotic symptoms or hallucinogen-persisting perceptual disorder, and no participants required psychiatric hospitalization [41,42,43].

Intranasal esketamine was generally well tolerated. The severity of most AEs was mild to moderate. The most common psychiatric AEs were dissociation, dissociative disorder, and transient anxiety, while the most common somatic AEs were headaches, dizziness and vertigo, nausea, dysgeusia, and elevated blood pressure. Most AEs were transient with onset shortly after dosing and resolution on the same day. Perceptual changes and dissociative symptoms began shortly after the start of intranasal dosing, peaked at approximately 30 to 40 min, and resolved by 1.5 to 2 h. Most patients were considered ready for discharge at 1.5 h after dosing. All studies revealed that serious AEs occurred in individuals, such as suicidal ideation, suicidal attempt, exacerbation of depressive symptoms, agitation, depersonalization, anxiety, disorientation, autonomic nervous system imbalance, hypothermia, lacunar stroke, sedation, simple partial seizures, and fractures. Some of them led to withdrawal [26,33,34,35,36,37,38,39,40]. Some patients needed a reduction of the esketamine doses due to intolerance [33,37]. No evidence of withdrawal symptoms was observed during 1 or 2 weeks after cessation of treatment [33,34,35,36,37,38,39,40]. There were no reports of drug abuse or cravings during the follow-up phase [33,34,35,36,37,38,39,40].

## 4. Discussion

The number of depression diagnoses is continuously increasing and the current therapies are not efficient enough. Many cases of TRD suggest that the common views on the origin of the disease are wrong [10]. While recent systematic umbrella review strongly suggested that so far the most popular serotonin hypothesis of depression may be unsubstantiated, there are other theories that need to be addressed (e.g., imbalance of other monoamines (noradrenaline, dopamine), hypothalamic–pituitary–adrenal axis changes, neuroinflammation, disturbed neurogenesis, genetics, epigenetics, and environmental factors) [44]. That being said, the need to find novel agents is an indisputable priority in psychiatry. So far, researchers have managed to demonstrate the antidepressant effect of a dissociative, esketamine, which led to its recent approval for TRD and MDD with suicidal thoughts or behaviors [15]. Currently, the interest revolves around the potential use of psilocybin, a serotonergic hallucinogen, in depression treatment. Table 8 briefly lists major features of both substances. Particular attention should be paid to the mechanism of action of both substances. Psilocybin acts on 5HT, especially 5HT2A, receptors, which are thought to be responsible for producing psychedelic effects [45]. Moreover, in fMRI studies, psilocybin decreased cerebral blood flow (CBF) in the amygdala and posterior cingulate cortex, hippocampal structures found to be unpaired in depression [19,46,47]. Psilocybin was also found to modulate the brain default mode network (DMN), a functional activity which, when abnormal, is linked with psychiatric disorders [48]. Esketamine produces antidepressant effects mainly by acting as N-methyl-D-aspartate glutamate receptor antagonist, which results in activating mTOR pathway and consequently enhancing neuroplasticity [49]. A recent study conducted by Wojtas et al. indicated that both ketamine and psilocybin increase dopamine, serotonin, glutamate, and GABA extracellular levels in the frontal cortex in rats. These results may support the hypothesis of mTOR signaling and neurogenesis as crucial factors in depression and explain the common antidepressant mechanisms of esketamine and psilocybin [50]. 

The major concern in both psilocybin and esketamine cases are the mind-altering effects they produce, as these are the reason for the recreational use of those substances. Esketamine can cause dissociation, the state of feeling disconnected from one’s thoughts, space, and time, which usually lasts about 40 to 90 min after administration [51,52]. Psilocybin produces an experience of an altered state of consciousness (ASC), commonly called hallucination, lasting 4 to 6 h [53]. While no correlation has been found between dissociation and esketamine’s antidepressant effect [54,55], ASC and mystical-type experiences that can occur during psilocybin sessions are considered to have profound and positive therapeutic outcomes [56]. The important issue that also needs to be addressed when regarding those substances is the risk of abuse and misuse [57,58]. Both esketamine and psilocybin have a history of recreational use which eventually led to their prohibition in the past. Although no drug-seeking behavior was noticed in esketamine clinical trials, it may have abuse potential and should only be administered under supervision and only to patients who are registered in the system [59]. Psilocybin is also a controlled substance and is known for being used recreationally. However, its abuse potential, as well as the risk of dependence, seems to be small, mainly due to long-lasting effects and immediate tolerance [11,60]. The risk of psilocybin off-label use can be restricted, similarly to esketamine’s, by administering the drug in a controlled environment. The psilocybin clinical trials are carefully prepared, and the rules of proper set and setting are followed; therefore, the same procedures can be applied in future treatment if the substance were to be approved [61]. In the case of psilocybin, a mind-altering state can be powerful and sometimes even overwhelming; thus, the presence of a qualified therapist both during and after the session is crucial for a safe and efficient treatment. Notably, psilocybin sessions should be considered as a part of psychotherapy rather than an individual experience [42]. Notably, the studies included in this review indicated no serious adverse events occurred during and after psilocybin treatment and no serious AE-related withdrawal was detected (one person discontinued because of vomiting, which was not classified as serious [41]). Concurrently, all esketamine studies noted serious adverse events which, in some cases, led to discontinuation [26,33,34,35,36,37,38,39,40]. Nevertheless, more psilocybin studies need to be conducted to verify its safety.

There are a few limitations of the studies included in this review. As esketamine is already FDA-approved, we decided to focus especially on psilocybin studies. As it is presented in Table 5, there is inconsistency within the studies. Outcomes were measured by using different scales, making it somewhat challenging to compare those studies to the esketamine ones (Table 2 and Table 3). The other limitation of the psilocybin studies is the small size of the samples, ranging from 29 to 59 (Table 1). Another limitation to consider is the integrity of blinding procedures, which is common for both esketamine and psilocybin. As these substances, especially psilocybin, are highly psychoactive, it may be impossible to maintain the double-blind design which is today’s standard for clinical studies. Researchers’ attempts to overcome this problem include administering active placebo, such as niacin [43], or low doses of the same psychoactive substance that is administered in higher doses to the experimental group [42].

## 5. Conclusions

As this review indicated, esketamine is rapid in its onset and its effects can persist for up to 7 weeks. Those effects are superior compared to the standard-of-care oral antidepressants. Moreover, esketamine reduces suicidal thoughts and severe depression symptoms. Psilocybin demonstrates both rapid and long-term antidepressive effects as well, with comparable effects to standard-of-care pharmacotherapy. Both esketamine and psilocybin were proven to be safe if used in controlled conditions. The studies suggest that psilocybin effects are comparable with the already approved esketamine and standard-of-care antidepressants and could be used as an antidepressant agent as well.

## Figures and Tables

**Figure 1 ijms-23-11450-f001:**
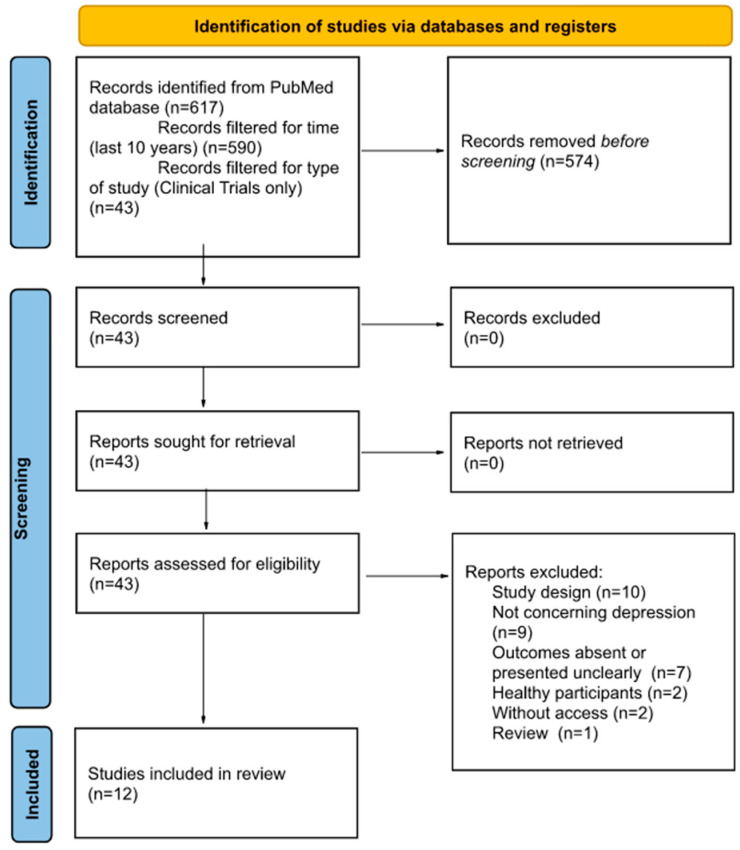
Flowchart presenting article selection process.

**Table 1 ijms-23-11450-t001:** Recent randomized, placebo-controlled studies concerning psilocybin and esketamine in depression.

Author, Year	Substance	Phase	No. Part.	Condition
Griffiths, R. 2016 [41]	Psilocybin	Phase 2	51	Depression and/or anxiety
Ross, S. 2016 [42]	Psilocybin	Early Phase 1	29	Depression and/or anxiety
Carhart-Harris, R. 2021 [43]	Psilocybin	Phase 2	59	Moderate-to-severe Major Depressive Disorder
Daly, E.J. 2018 [26]	Esketamine	Phase 2	57	Treatment Resistant depression
Canuso, C.M. 2018 [33]	Esketamine	Phase 2	68	Treatment Resistant Depression with Suicide Ideation
Fedgchin, M. 2019 [34]	Esketamine	Phase 3	346	Treatment Resistant Depression
Popova, V. 2019 [35]	Esketamine	Phase 3	227	Treatment Resistant depression
Daly, E.J. 2019 [36]	Esketamine	Phase 3	297	Treatment Resistant depression
Fu, D-J. 2020 [37]	Esketamine	Phase 3	226	Treatment Resistant Depression with Suicide Ideation
Ochs-Ross, R. 2020 [38]	Esketamine	Phase 3	51	Treatment Resistant Depression
Takahashi, N. 2021 [39]	Esketamine	Phase 3b	202	Treatment Resistant Depression
Ionescu, D.F. 2021 [40]	Esketamine	Phase 3	230	Treatment Resistant Depression with Suicide Ideation

**Table 2 ijms-23-11450-t002:** Esketamine rapid onset. Change in MADRS score at 2–4 h and 24 h after esketamine intake.

Change in MADRS after:	LS and MD from Baseline and AD + Placebo	Author, Year							
		Daly, E.J. 2018 [26] [a]		Canuso, C.M. 2018 [33] [b]	Fedgchin, M. 2019 [34] [a]		Popova, V. 2019 [35] [c]	Fu, D.-J. 2020 [37] [b]	Ionescu, D.F. 2021 [40] [b]
2–4 h postdose	MD from baseline	−14.3 [a1]	−17.6 * [a2]	−13.4 *	-	-	-	-	-
	LS MD from AD + placebo	−4.6 [a1]	−7.9 * [a2]	−5.3 *	-	-	-	-	−4.2 *
24 h postdose	MD from baseline	−15.7 * [a1]	−16.4 * [a2]	-	-	-	-	−16.4 *	−15.7 *
	LS MD from AD + placebo	−10.0 * [a1]	−10.7 * [a2]	−7.2 *	−3.0 ** [a1]	−2.2 ** [a2]	−3.3	−3.8 *	−3.9 *

Abbreviations: AD—(standard-of-care) antidepressant; LS—least square; MD—mean difference. [a]—results for 56 mg [a1] and 84 mg [a2], respectively. [b]—results for 84 mg. [c]—flexible doses, ranging from 56 mg to 84 mg. * statistically significant. ** statistical significance was not assessed as the primary endpoint was not met.

**Table 3 ijms-23-11450-t003:** Change in MADRS score at 25th-28th day of esketamine treatment.

		Author, Year							
		Canuso, C.M. 2018 [33] [a]	Fedgchin, M. 2019 [34] [b]		Popova, V. 2019 [35] [c]	Ochs-Ross, R. 2020 [38] [c]	Takahashi, N. 2021 [39] [b]		Ionescu, D.F. 2021 [40] [a]
Day 25.–28.	MD from baseline	-	−19.0 **	−18.8 **	−21.4 *	−10.0	−14.5 [b1]	−15.1 [b2]	-
	LS mean difference from AD + placebo	−4.5	−4.1 ** [b1]	−3.2 ** [b2]	−4.0 *	−3.6	0.6 [b1]	−0.9 [b2]	−3.7*

Abbreviations: AD—(standard-of-care) antidepressant; LS—least square; MD—mean difference. [a]—results for 84 mg. [b]—results for 56 [b1] and 84 [b2] mg, respectively. [c]—flexible doses, ranging from 56 to 84 mg. * statistically significant. ** statistical significance was not assessed as the primary endpoint was not met.

**Table 4 ijms-23-11450-t004:** Response and remission rates compiled at 4–7-week follow-up (time frames vary over different studies).

Author, Year	Substance	Scale	Response Rate (vs. Control)	Remission Rate (vs. Control)
Griffiths, R. 2016 [41]	Psilocybin	GRID-HAMD-17	92% (vs. 32%)	60% (vs. 16%)
Ross, S. 2016 [42]	Psilocybin	BDI	~80% (vs. ~15%)	~80% (vs. ~15%)
		HADS Depression	~70% (vs. ~40%)	~70% (vs. ~40%)
Carhart-Harris, R. 2021 [43]	Psilocybin	QIDS-SR-16	70% (vs. 48%) [a]	57% (vs. 28%) [a]
Daly, E.J. 2018 [26]	Esketamine	MADRS	56% [b]	42% [b]
Fedgchin, M. 2019 [34]	Esketamine	MADRS	54.1% and 53.1% (vs. 38.9%) [c]	36.0% and 38.8% (vs. 30.6%) [c]
Popova, V. 2019 [35]	Esketamine	MADRS	69.3% (vs. 52.0%)	52.5% (vs. 31.0%)
Ochs-Ross, R. 2020 [38]	Esketamine	MADRS	27.0% (vs. 13.3%)	17.5% (vs. 6.7%)
Ionescu, D.F. 2021 [40]	Esketamine	MADRS	59% (vs. 48.0%)	43.0% (vs. 27.0%)

Clinical response was defined as ≥50% decrease in measure relative to baseline; symptom remission was defined as ≥50% decrease in measure relative to baseline and a score of ≤7 on GRID-HAMD-17, HADS D ≤ 7, BDI ≤ 12, or MADRS ≤ 10. [a] Control group was administered escitalopram instead of psilocybin. [b] After open label phase, where all participants received esketamine. [c] Results for esketamine 56, 84 mg, and placebo, respectively.

**Table 5 ijms-23-11450-t005:** Antidepressant effects of psilocybin.

Author, Year	Dose	Time Frame	Primary Outcome Measure	Results	Secondary Outcome Measure	Results
Griffiths, R. 2016 [41]	22 or 30 mg/70 kg	5 weeks after the 1st session	GRID-HAMD-17	Significant difference between the Psilocybin-1st(Placebo-2nd) and Placebo-1st(Psilocybin-2nd) groups	BDI, HADS	Decrease, significant difference between the Psilocybin-1st(Placebo-2nd) and Placebo-1st(Psilocybin-2nd) groups
		5 weeks after the 2nd session = crossover		No significant difference between the Psilocybin-1st(Placebo-2nd) and Placebo-1st(Psilocybin-2nd) groups *		Decrease, no significant difference between the Psilocybin-1st(Placebo-2nd) and Placebo-1st(Psilocybin-2nd) groups *
		6-months’ follow-up		Significant difference from baseline		Decrease, significant difference between baseline and 6-months’ follow-up
Ross, S. 2016 [42]	0.3 mg/kg	6 weeks after the 1st session	HADS D, BDI	Significant difference from baseline in the Psilocybin-1st(Placebo-2nd) group. Significant difference between the Psilocybin-1st(Placebo-2nd) and Placebo-1st(Psilocybin-2nd) groups.	-	-
		6 weeks after the 2nd session = crossover		Significant difference from baseline in the Psilocybin-1st(Placebo-2nd) group. Significant difference from baseline in the Placebo-1st(Psilocybin-2nd) group in BDI, but not in HADS; No significant difference between groups in BDI, but not HADS *		
		26-week follow-up		Significant difference from baseline in both Psilocybin-1st(Placebo-2nd) and Placebo-1st(Psilocybin-2nd) groups; No significant difference between groups *		
Carhart-Harris, R. 2021 [43]	25 mg	6 weeks	QIDS-SR-16	No significant difference between the psilocybin and escitalopram groups	HAMD-D-17, MADRS, BDI	As the primary endpoint was not met, the secondary outcomes’ significance was not adjusted, but in general the differences between groups favored psilocybin over escitalopram

Only depression-related outcomes were included in this table. * After the 2nd session, both groups received psilocybin. No difference means that score decreased in Placebo-1st (Psilocybin-2nd) group, while score in Psilocybin-1st (Placebo-2nd) group was sustained.

**Table 6 ijms-23-11450-t006:** The most common adverse effects during esketamine treatment.

Author, Year	No. of Assessed Patients	Dissociation [%]	Headaches [%]	Dizziness [%]	Vertigo [%]	Nausea [%]	Dysgeusia [%]	Elevation in BP [%]
Canuso, C.M. 2018 [33]	35 (DB), 27 (FU)	31.4 (DB), 0 (FU)	31.4 (DB), 7.4 (FU)	34.3 (DB), 3.7 (FU)	11.4 (DB), 0 (FU)	37.1 (DB), 0 (FU)	31.4 (DB), 3.7 (FU)	N/A
Daly, E.J. 2018 [26]	56 (DB), 57 (OL)	20 (DB)	21 (DB), 14 (OL)	36 (DB), 39 (OL)	7 (DB)	18 (DB), 16 (OL)	18 (DB), 23 (OL)	Hypertension-5 (DB)
Fedgchin, M. 2019 [34]	231	26.8	20.3	25.1	20.8	29.4	16.0	8.2
Popova, V. 2019 [35]	116	26.1	20.0	20.9	26.1	26.1	24.3	9.6
Daly, E.J. 2019 [36]	152	23.0	17.8	20.4	25.0	16.4	27.0	6.6
Fu, D-J. 2020 [37]	113	29.2	18.6	35.4	6.2	20.4	14.2	16.8
Ochs-Ross, R. 2020 [38]	72	12.5	12.5	20.8	11.1	18.1	5.6	12.5
Takahashi, N. 2021 [39]	122	37.7	12.3	36.1	15.6	18.0	N/A	41.0
Ionescu, D.F. 2021 [40]	114	38.6	21.9	41.2	6.1	33.3	25.4	6.1

Abbreviation: N/A—not applicable; DB—double-blind; FU—follow-up; OL—open label.

**Table 7 ijms-23-11450-t007:** Adverse effects by type occurred in more than 5% of patients during psilocybin treatment.

Author, Year	Number of Assessed Patients	Dose	Psychiatric	Neurological	Cardiovascular	Gastroenterological	General
Griffiths, R. 2016 [41]	50 cross-over	(high dose 22 or 30 mg/70 kg), (low dose 1 or 3 mg/70 kg)	psychological discomfort: 32 (high dose), 12 (low dose); transient anxiety: 26 (high dose), 15 (low dose)	-	elevation in SBP: 34 (high dose), 17 (low dose); DBP: 12 (high dose), 2 (low dose)	nausea/vomiting: 15 (high dose session), 0 (low dose)	physical discomfort: 21 (high dose session), 8 (low dose session)
Ross, S. 2016 [42]	28 cross-over	0.3 mg/kg	transient anxiety: 17; transient psychotic-like symptoms: 7	headaches/migraine: 28	elevation in BP and HR: 76	nausea: 14	-
Carhart-Harris, R. 2021 [43]	30 (6-week trial period), 30 (dosing-day 1)	25 mg	feeling jittery: 7 (6-week trial), 0 (dosing-day 1)	headaches: 67 (6-week trial), 43 (dosing-day 1); migraine: 10 (6-week trial), 0 (dosing-day 1); dizziness: 7 (6-week trial)	-	nausea: 27 (6-week trial), 13 (dosing-day 1); vomiting: 7 (6-week trial), 0 (dosing-day 1	fatigue: 7 (6-week trial), 0 (dosing-day 1)

**Table 8 ijms-23-11450-t008:** Summary of psilocybin and esketamine features.

Substance	Psilocybin	Esketamine
Mechanisms of action	Activates the 5-HT2A receptors	Blocks subsets of NMDA receptors on GABA interneurons
Conjunction	In conjunction with psychotherapy	In conjunction with SSRI, SNRI
Depression tupe	Under research in Major Depressive Disorder and Depression Related to Life-Threatening Diseases	Treatment-Resistant Depression, Major Depressive Disorder with Suicidal Thoughts or Behaviors
Influence on cognitive functions	Mind-altering effects, “hallucinations”	Dissociation
Influence on suicidal thoughts	Likely reducing	Reducing
Advantages	Neither addictive nor hepatotoxic, and not toxic to tissues; Route of administration: oral; Rapid onset and long-term effects (up to 6 months) Seems to produce no serious adverse events Probably no or little abuse potential	Route of administration: nasal; Rapid onset and long-term effects (up to 7 weeks)
Limitations	Mind-altering side-effects, misuse potential (must be administered under monitored conditions) Caution: may increase the risk of mania in patients with bipolar disorder, and may increase the risk of cardiac arrest and death in patients with cardiovascular diseases	Mind-altering side effects (administration must take place in a clinic and be closely monitored) Not recommended for use during pregnancy or in women of age, not applicable to children under 7 years of age Reproductive Contraindications: hypersensitivity (also to ketamine), aneurysm, intracerebral hemorrhage, recent heart attack Can have abuse potential
Adverse Events	Minor side effects: transient increase in blood pressure, body tremors, fear and sadness, mild to moderate transient headache	Can produce serious adverse events (e.g., suicidal ideation, suicidal attempt, lacunar stroke, seizures) Other: sedation, transient increase in blood pressure

## Data Availability

Not applicable.

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
