# Peer review of "Esketamine and Psilocybin—The Comparison of Two Mind-Altering Agents in Depression Treatment: Systematic Review"

_ijms, 2022, doi:10.3390/ijms231911450_

Round 1

Reviewer 1 Report

In the paper, the authors reviewed and compared the antidepressant effects of esketamine and psilocibin in patients with depression. The analysis was based on 12 clinical trials, 3 of which were for psilocybin and 9 for esketamine. The issue is very relevant, as interest in the use of psychedelics in the treatment of depression has been experiencing a renaissance in recent years. The more disappointing was the implementation of the topic undertaken by the authors. As the authors noted, esketamine is authorized for use in the United States but also in the European Union, and its therapeutic effects, usage and safety profile were reviewed many times (54 records in the the last year in PubMed database, search terms: esketamine AND depression; filtered by article type: review), thus another review seems redundant. Next, the discussion of the effects of psilocybin was carried out on the basis of only three studies whose results are inconsistent. This makes the comparison of the properties of psilocybin and ketamine unreliable. In my opinion, it would make more sense to review clinical trials with various psychedelics (like: psylocibin, LSD, MDMA) in depressed patients and discuss their potential advantages and disadvantages compared to esketamine as an authorized drug.

Minor concerns:

-        The exact dosage of drugs in each clinical trial should be presented in the tables. Now it is difficult to know whether the drugs have been used once or repeatedly and in what doses.

-        The manuscript should be checked for editing and punctuation errors.

-        Table captions should be checked; for example, in Table 4 references [a] and [b] are missing.

-        The formatting of references should be improved according to the MDPI Reference List and Citations Style Guide.

Author Response

Dear Reviewer,

Thank you for bringing these concerns to our attention. 

As esketamine has already been approved in depression treatment it may seem unnecessary to review it as an antidepressant agent, however we found it needful in order to compare its effects to those produced by psilocybin. We agree that, using current research, the comparison was rather difficult to draw, as the psilocybin studies varied from each other in design. Moreover, scales used in psilocybin studies and esketamine studies were different, which made this comparison challenging as well. We were aware of these limitations and mentioned them in Discussion. Nonetheless, due to inclusion (only studies which enrolled patients with depression; randomized, placebo controlled trials) and exclusion (healthy subjects) criteria we had implied in this review only three psilocybin studies could be included. Studies concerning other psychedelic agents such as LSD, DMT or non-classical psychedelic MDMA did not meet these criteria, thus could not be included in this paper. To our knowledge our paper is the first attempt to compare esketamine and psilocybin clinical effects in depression, nevertheless we agree that conducting a review using different methods in the future may give relevant results. We will imply your suggestions in the paper in the best possible way. 

Kind regards

Dominika Psiuk

Reviewer 2 Report

The present work, performed by Dominika Psiuk, Emilia Nowak, Natalia Dycha, Urszula Łopuszańska, Jacek Kurzepa and Marzena Samardakiewicz review the effects of two psychoactive substances: psilocybin and esketamine, for a rapid actions on the treatment for depressive symptoms and suicidal ideation.

The subject is interesting and highly relevant because Major Depression Disorder (MDD) is increasing worldwide, and worryingly in young populations: children and adolescents, being suicide a very frequent outcome.

The work is well documented and clear presented.

Comments for the manuscript:

Page 2, lines 67-71: It would be beneficial for the reader mention additional hypothesis for depression, including those in which other signaling systems participate: glutamate, GABA, cholinergic.

Introduction:

The number of some references along the text does not correspond with those in the list of references. This fail started with reference 16.

Results:

Table 4: This reviewer does not detect where are [a]:control group was administered escitalopram instead of psilocybin” and [b]: “after open label phase, where all participants received esketamine”.

Page 2, lines 238, 239: some lines above (227, 228) it is stated that suicidal ideation or behavior were key exclusion criteria in references 34 and 36; whereas in lines 237, 238 these criteria are included in these references. One possible problem here is also that the number of some references in the text do not match with those in the list of references.

Discussion:

Regarding Table 8: a) Why, the two compounds, psilocybin and esketamine, with two different sites and mechanisms of action, have similar effects in alleviating symptoms of depression? b) What advantages would have psilocybin compared with esketamine?

Authors may include some lines speculating these questions.

Author Response

Dear Reviewer,

Thank you for your suggestions. We will apply your suggestions in the paper in the best possible way.

We would like to explain one of your concerns about suicidal behavior that occured in studies. Three studies enrolled patients specifically with suicide ideation or behavior, others excluded patients with such symptoms. The fragment in lines 237-239 refers to all studies, as suicidal symptoms occured in some patients during esketamine treatment regardless of whether they were initially suicidal or not. We hope that our explanation is sufficient and clear.

Kind regards

Dominika Psiuk

Reviewer 3 Report

The study is well executed, designed and reported. The topic is highly relevant regarding the treatment of Depression. The research problem is properly formulated, and the materials, methods and the aims are stated clearly. The research design is applicable, and the results are explained well. The conclusion is clear and the recommendations reflect the aims of the study. However, there some minor corrections to be done.

The following minor corrections should be made:

1) Thirteen(13) of the references are older than 5 years. I want to recommend to the authors to revise these references and try to get more recent references younger than 5 years. I understand that some of these references are primary references and needs to be included. 

2) Some of the references are numbered incorrectly. For example reference 15 is marked 15 and 16. The same goes for references 17,23,27 and 30. Please correct

Author Response

Dear Reviewer,

Thank you for your suggestions. 

We agree that some of cited papers are old, however in most cases they regard information that is not likely to change within years (for example SSRIs’ and other agents’ pharmacodynamics). We will try to apply your suggestions to the paper and change literature in other cases.

Kind regards

Dominika Psiuk

Round 2

Reviewer 1 Report

The authors took into account only my minor concerns. I still believe that the manuscript adds little to current knowledge about the potential use of psychedelics in the treatment of depression, but I accept the authors' explanations.

Author Response

Dear Reviewer,

Thank you again for sharing your suggestions. We will undoubtedly take them under consideration again when composing another work in this field.

Kind regards

Dominika Psiuk